# Custom-Made Foot Orthoses Reduce Pain and Fatigue in Patients with Ehlers-Danlos Syndrome. A Pilot Study

**DOI:** 10.3390/ijerph17041359

**Published:** 2020-02-20

**Authors:** María Reina-Bueno, Carmen Vázquez-Bautista, Inmaculada C. Palomo-Toucedo, Gabriel Domínguez-Maldonado, José Manuel Castillo-López, Pedro V. Munuera-Martínez

**Affiliations:** Department of Podiatry, University of Seville, 41009 Seville, Spain; carmenzaz@us.es (C.V.-B.); ipalomo@us.es (I.C.P.-T.); gdominguez@us.es (G.D.-M.); jmcastillo@us.es (J.M.C.-L.);

**Keywords:** Ehlers-Danlos syndrome, custom-made foot orthoses, foot pain, quality of life, fatigue

## Abstract

Background: Pain and fatigue are major clinical manifestations in patients with Ehlers-Danlos Syndrome (EDS). The aim of this study is to measure change of the effects of custom-made foot orthotics on some manifestations related to EDS, such as foot pain, foot functionality, fatigue, and quality of life. Methods: Thirty-six patients with EDS wore foot orthoses for three months. Foot pain, foot-related disability, foot functionality, fatigue, and quality of life were measured using the 11-point Numeric Rating Scale, the Manchester Foot Pain and Disability Index, the Foot Function Index, the Fatigue Severity Score, and the 12-Item Short Form Health Survey questionnaires, respectively, at the beginning and after 3 months. Results: Participants demonstrated significantly improved foot pain (*p* = 0.002), disability related to foot pain (*p* < 0.001), foot functionality (*p* = 0.001), fatigue (*p* < 0.007), and mental health-related quality of life (*p* = 0.016). The physical health-related quality of life did not show significant changes. Conclusions: The use of custom-made foot orthoses help in the management of the symptoms by participants. This study could contribute to the foot specialists being considered as an additional member in multidisciplinary teams that are trying to develop an approach for patients with EDS.

## 1. Introduction

The Ehlers-Danlos syndrome (EDS) is a heterogeneous group of heritable disorders affecting the matrix of proteins of connective tissue [1]. The prevalence is estimated between 1:5000 and 1:20,000 [2], and is more frequent in females, representing 90% of all cases [3].

The 2017 international classification of the EDSs recognised 13 subtypes [4], and foot pathology was present in several of them. The hypermobility type is the most common variety, with an estimated prevalence of 80–90% of all cases. Its classical symptoms are hyperextensible skin, joint hypermobility frequently associated with limb pain and joint dislocations, and blood vessel and tissue fragility [5].

One of the major clinical manifestations in patients with EDS is pain, which may be associated with disability, depression, anxiety, or physical and mental health-related quality of life impairment. [6]. Fatigue has also been identified as one of the most debilitating symptoms in EDS, and is highly related to sleep disturbances, concentration problems, poor social functioning, and pain severity [7].

Lower limb problems are considered common manifestations in patients with EDS. Some of these problems may be related to the feet. Previous research has been done on foot type analysis in individuals with the EDS hypermobility type, and found that 45% have cavus feet, 27.5% planus feet, and 27.5% normal arch. [8] In a qualitative study, it was shown that foot pain was an important complaint among the participants [9]. Several authors have reported about foot pain in EDS, and different foot problems other than pain, like for example, stress concentration in the forefoot that may perturb standing and walking and cause pain that is frequently reported by people with EDS [10]. In addition, gait patterns have been observed to be altered in people with EDS, as they have been observed to show a higher range of centre of pressure displacement in the antero-posterior and medial-lateral direction, a higher trajectory length of centre of pressure values, and a severe postural instability as compared to controls [3].

Treating foot pathology is challenging due to the lack of evidence-based studies on the effectiveness of treatments for patients with EDS, as published evidence on conservative treatment is limited to papers on small cohorts or single case design. Previous research has been done on physical therapies for lower limb symptoms in children diagnosed of EDS, with unclear results [11]. Specifically, regarding orthopedic management of EDS, some recommendations have been reported, such as splints or orthotics/arch supports [12,13], but no clinical studies exist that address the effectiveness of orthopaedic treatment for foot problems. Therefore, the aim of this study was to describe the effects of custom-made foot orthotics on some EDS-related manifestations, such as foot pain, foot functionality, chronic fatigue, and quality of life.

## 2. Materials and Methods 

An observational, pre-test post-test study was carried out. This study was authorised by the Clinical Podiatric Area of the University of Seville and by the Research Ethical Committee of the University Hospitals Virgen Macarena and Virgen del Rocío. Written consent was obtained from all the participants.

A priori sample size was calculated to detect changes with an unknown effect size, so a medium (neither small nor large) effect size was chosen (Rosenthal’s r or Cohen’s d between 0.5 and 0.8), for a contrast of pre- and post-test measurements on the same sample (matched pairs), assuming type I and type II errors of 0.05 and 0.1 (Power = 1 − β = 0.90), respectively. With these design values, the G*Power 3.1.9.4 software package (Universität Düsseldorf, Germany) was used to calculate the minimum sample size, which was found to be 36 cases. Loss to follow-up was predicted to be 10%, so a minimum of 40 participants was required.

A Spanish national organisation was contacted to find participants (Joint Hypermobility, Collagenopathies and Ehlers-Danlos Syndromes National Association). Data recording was carried out in various centres in Spain, both public and private (the Podiatry Clinical Area of the University of Seville, the Podiatry Clinic of the University of A Coruña, a private outpatients centre in Jaen, and the Podiatric Hospital of the University of Barcelona), from October 2017 to October 2018.

Participants for this study had to be adults with EDS diagnosed by a medical specialist, and with self-reported foot symptoms. Those with a neurologic disease, cognitive impairment, or difficulties for an independent gait, or those already wearing foot orthoses, who had undergone previous foot surgery, recently commenced pain medication, or were on regular pain management medication, were excluded.

Clinical and demographic data were collected, including age, gender, weight, height, and years for diagnosis. Patients with EDS were explored and Foot Posture Index (FPI) (a validated method for quantifying standing foot posture) [14], Beighton’s criteria [15], and Manchester scale for hallux valgus were recorded for both the right and left foot [16].

Weight-bearing phenolic foam moulds of the feet were obtained to make the custom-made foot orthoses. They consisted of a polypropylene layer of 2 mm from heel to just proximal to the metatarsal heads, and an upper sheet of 40 Shore A, 3-mm thick polyethylene foam (Figure 1). 

The participants received instructions for the utilization of the foot orthotics for at least 7 h a day for 3 months in their usual footwear. Previously, the researchers had verified that the foot orthoses fitted properly to the foot and shoe, and that they did not cause discomfort to the participants. Once a month, the participants were contacted by phone by one of two authors (MRB or CVB) to make sure that the participants were following those instructions and to remind them about the importance of wearing the orthotics during the follow-up period.

The day the participants received the foot orthoses; foot pain, foot functionality, disability related to foot pain, intensity of fatigue, and quality of life were recorded.

### 2.1. Pain Measures 

Self-reported pain intensity over the last month was assessed with an 11-point numeric pain rating scale (NPRS) with 0 = no pain to 10 = pain as bad as it can be. Pain days were also recorded as the number of days on which the patient felt foot pain in the previous week by assigning a whole number between 0 and 7.

### 2.2. Functional Measures

Foot functionality was measured using the foot function index (FFI) [17]. This questionnaire is validated for the Spanish population. This is a questionnaire with 23 items that are divided into three domains: Foot pain, disability, and functional limitation. The values range from 0 and 100, with higher values corresponding to greater pain, disability, and limitation.

### 2.3. Disability Measures

Disability related to foot pain was measured using the Manchester Foot Pain and Disability Index (MFPDI) [18]. The values of this index range from 0 to 38, with higher values corresponding to greater disability.

### 2.4. Fatigue Measures

To evaluate the intensity of fatigue, the participants were asked to complete the fatigue severity scale (FSS) [19], which has previously been used in studies with individuals affected by the disorder [20]. The FSS consists of 9 items with a 7-point Likert-type response format indicating the degree of agreement with each statement. The scoring is done by calculating the average response to the questions (adding up all the answers and dividing by nine). Higher scores mean greater fatigue severity. 

### 2.5. Quality of Life Measures

The SF-12 questionnaire was used to collect data about the quality of life [21]. This uses values between 0 and 100, with higher values corresponding to lower quality of life.

These data were collected again after the 3-month follow-up period.

### 2.6. Data Analysis

The analysis of the data was carried out using the statistical software IBM SPSS Statistics 22^®^ (IBM, Armonk, NY, USA). The descriptive data provided the mean values and the standard deviations, medians, and the absolute frequencies and percentages depending on whether the variables were continuous or categorical. Normality tests were conducted for the inferential analysis to determine the most appropriate test to use. Data showed an abnormal distribution, so non-parametric tests were carried out (Wilcoxon signed-rank test). The effect size was calculated using Rosenthal’s r to analyse the magnitude of the differences. The differences were classified according to the following for both parameters: Below 0.2, no effect; 0.2–0.5, small effect; 0.5−0.8, medium effect; above 0.8, large effect [22]. The confidence level set priori was 95%.

## 3. Results

Forty-one people with an EDS diagnosis initially participated in the study (36 women, 87.8%). Five of them were excluded from the statistical analysis because they did not attend the check-up after the 3-month follow-up period. One participant abandoned the study because of discomfort about the treatment, two because of inappropriate footwear, one because of an acute new motor disease, and the last one was impossible to contact after a 2-month follow-up period. Therefore, the final sample consisted of 36 participants (32 women, 88.9%) with a mean age of 39.6 ± 12.4 years, BMI of 24.3 ± 5.0, and mean time from diagnosis of 2 ± 1.8 years. Other participants’ self-reported general characteristics are show in Table 1 and Table 2. 

Self-reported and observed foot problems are summarised in Table 3.

The plantar orthoses improved some evaluated variables after 3 months of use: Foot pain (evaluated using 11-NRS, days with pain per week, and FFI scale), disability (FFI and MFPDI), activity restriction (FFI test), quality of life (mental status) (SF-12 scale), fatigue (FSS), and total FFI. The quality of life (physical function) (SF-12 scale) did not show significant improvement. The values of these variables at the beginning and end of the follow-up period, as well as the *p* values and size effect, are shown in Table 4.

The observed improvement in disability from 49.4 points to 35 points was measured in FFI, together with the improvement in functional limitation from 20 points to 6 points in FFI and from 14 to 11 in MFPDI, shows an improvement in patient mobility after 3 months with the orthoses. 

It was observed that the effect of the orthoses on foot health was evidenced by the improvement both in the FFI and MFPDI. The FFI decreased by 12 points or more in 50% of the patients and the MFPDI decreased from 23 initial points to 20 points at the end of the intervention (*p* < 0.001).

## 4. Discussion

This study was designed to describe the effects of custom-made foot orthotics on some important determinants affecting the daily lives of patients with EDS, such as pain (specifically foot pain) or fatigue. To the authors’ knowledge, this is the first study that reports benefits of treating EDS patients with foot orthotics in relation to foot pain, foot functionality, fatigue reduction, and partial improvement of quality of life. Other studies explore the effectiveness of other physical therapies with limited benefits in children with hypermobility [11]. 

Musculoskeletal pain has been described as one of the major determinants for the deterioration of quality of life [20]. Abnormal foot function may contribute to foot and lower limb discomfort and pain that are likely to limit the willingness of patients with EDS to walk, leading to a reduction in physical activity and muscular weakness, with the final result that the quality of life in these individuals may be significantly affected [10]. Treating foot pathology is challenging due to the lack of evidence-based studies on the effectiveness of treatments for patients with EDS feet. Mechanical therapies may contribute to increase joint proprioception control, reduce postural instability, and improve motor coordination due to hiperlaxity. Custom-made foot orthoses have been demonstrated to reduce foot pain in other chronic diseases, such as rheumatoid arthritis [23], but not in EDS. However, one study found that pain and its impact on daily life were significantly worse in patients with EDS-Hypermobile type than in those with rheumatoid arthritis [24]. 

Research regarding orthopaedic instruments other than custom-made foot orthotics (i.e., splints) for treating foot problems in people with EDS is limited. Branson et al. published their pain management strategy in an adolescent girl with EDS. They used physical therapy, oral medication, and psychological therapy, and reported that the use of an AFO (Ankle Foot Orthosis) splint contributed to ankle stabilisation and diminished the fall risk [25]. 

Arthur et al. obtained 1179 respondents in their study with anonymous surveys [6], by which self-reported methods for pain management were evaluated in patients with EDS, and found that the most successful therapies were opioids, massages, splints, therapy with heat, and avoiding dangerous activities. In addition, the use of braces for short periods has been reported to improve joint stability in the case of recurrent sprain.

According to our results, foot pain was diminished by two points (median 6 to 4) after 3 months use of foot orthotics. It must be pointed out that a change of one point or more in the NPRS has been shown to be a minimal clinically important change [26]. Moreover, self-reported pain measured with FFI (median 69.3 to 45), or the number of days with pain in the last week (median 7 to 2.5), also showed a significant reduction after the follow-up (*p* < 0.001 and *p* < 0.002, respectively).

Among conservative approaches, physiotherapy is considered a successful strategy [27,28,29,30,31,32]. There is some evidence that EDS improves with exercise, but there are no clear cause–effect relationships for exercise [33]. Physiotherapy could be supported by foot orthotics in the treatment of major and minor complaints in these patients.

Research focusing on features other than pain across all subtypes of EDS has identified fatigue as one of the most debilitating symptoms reported by patients. In our study initial fatigue measured by FSS was 45.19 ± 12.45 (mean standard ± deviation, median = 51), and after 3 months use of custom-made foot orthoses was 40.98 ± 14.47 (mean standard ± deviation, median = 45.5), although the size effect was small (*r* = 0.45). One must be cautious when interpreting these results, as it is well known that fatigue in EDS is possibly influenced by several factors, such as poor sleep quality [34], kinesiophobia [35], orthostatic intolerance [36], or low physical activity participation [37]. Fatigue has been associated with greater pain and functional impairment [7,34], so one could think that it is important to consider treating pain when attempting to treat fatigue [36]. Given the results of the current study, new directions for the improvement of the management of people affected by EDS are highlighted. 

Chronic fatigue in EDS-HT includes bodily and mental fatigue [38]. Living with daily physical pain may limit participation in social life [9]. Patients suffering from chronic pain may experience an increase in their disability or there could be an effect on certain psychosocial life aspects, such as affecting mood [39], influencing social relationships [34], or favouring anxiety or depression [5], which may finally lead to psychological problems. According to the results obtained by the SF-12 questionnaire, the mental component was worse than that of the reference Spanish population [21] but was significantly higher compared to the initial situation after the follow-up (median 38.55 vs. 32.85, respectively, *p* = 0.016). However, the size effect was small (*r* = 0.40). On the other hand, this change was not significant in the physical component, which only increased by 2 points. Improving this mental dimension may have been a result of foot pain reduction and foot functionality improvement, with a consequent enhanced sense of well-being. Nevertheless, the possibility that participation in the study could have had a positive effect on the individuals’ outlook must be considered, as patients with EDS have difficulty when they try to gain access to appropriate treatment [1,6].

These findings should not be extrapolated to all people with EDS because this study has certain limitations. The patients were contacted via a national organisation and freely invited to be included, hence it is likely that those who had more severe foot problems decided to participate and might have more severe symptoms than a random sample of people with EDS. Patients with only mild symptoms might not seek medical attention and might not be interested in the study, or may not have even been diagnosed and have joined patients’ organisations.

Another limitation may be the relatively small sample size and not having included a control group. EDS is a rare condition and large experimental or more experimental groups are difficult to bring together. Some of the authors have previously reported the effectiveness of custom-made foot orthoses in other painful conditions by means of clinical trials with control groups (placebo) or a systematic review, and the benefit of wearing the orthoses was evidenced [23,40,41].

It was not taken into account the potential effect of pain medication on the participants, as this is a common practice among patients with EDS. Despite that, the authors think that these results are clinically important, because those participants who had recently commenced pain medication or were on regular pain management medication were excluded, and because even with the use of sporadic pain alleviating medication, self-reported pain was high at the beginning of the study, and foot orthoses contribute to significantly diminishing pain.

The EDS is recognised as a severely disabling disease, with chronic pain and fatigue as important factors related to severe deterioration of quality of life. Clinical studies are necessary to improve the general understanding of EDS and help symptoms management and disability prevention. Some papers mention the convenience of complex musculoskeletal problems being addressed by a multidisciplinary team approach, including a variety of specialists, but none of them include podiatrists [12,42]. This study could contribute to the inclusion of foot specialists in multidisciplinary teams who can try a new approach for patients with EDS.

## 5. Conclusions

The use of custom-made foot orthoses for 3 months improved foot pain disability related to foot pain, foot functionality, fatigue, and mental health-related quality of life in the patients with EDS who participated in this study. The physical health-related quality of life did not show significant changes.

The results of this study could inform future larger randomised controlled trials. 

## Figures and Tables

**Figure 1 ijerph-17-01359-f001:**
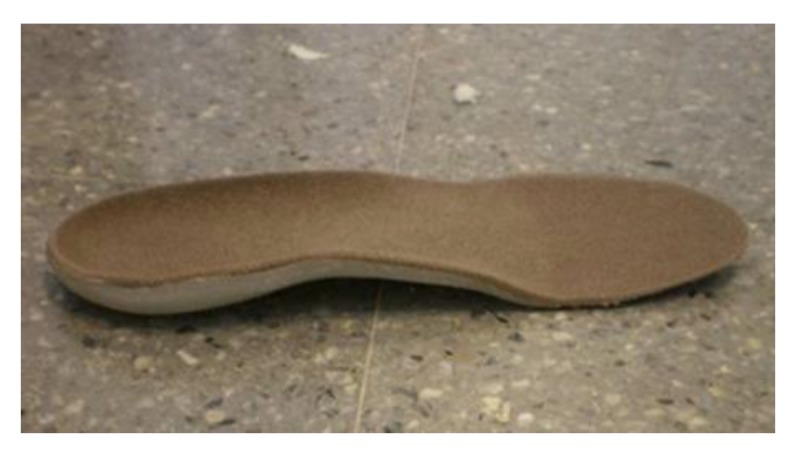
Type of orthotic device administered.

**Table 1 ijerph-17-01359-t001:** Participants’ comorbidities.

Participants’ Comorbities	N	Percentage (%)
Patient’s familiar history of EDS	19	52.8
Arterial hypertension	6	16.7
Diabetes Mellitus	2	5.6
Previous falls	22	61.1
Allergies	22	61.1
Previous surgery	30	83.3
Current smoker	14	38.9

**Table 2 ijerph-17-01359-t002:** Participants’ characteristics.

Participants’ Charateristics	N	Percentage (%)
Type of Ehlers-Danlos Syndrome Classic	3	8.3
Hypermobile	26	72.2
Vascular	3	8.3
Unknown	4	11.1
Abnormal scarring	21	58.3
Frequent hematomas	31	86.1
General joint pain	33	91.7
Low back pain	30	83.3
With some degree of disability	12	33.3
Disability grade (Mean ± SD)	45.8 ± 30.4	

**Table 3 ijerph-17-01359-t003:** Participants’ foot problems.

	N	Percentage (%)
Previous frequent sprains	31	86.1
Callus elimination	11	30.6
Previous foot orthotics	19	52.8
Previous physical therapy	12	33.3
Previous foot surgery	4	11.1
Papules	18 (5 painful)	50.0 (13.9)
Toe deformities	21	58.3
HAV (right foot)		
Mild	9	25.0
Moderate	5	13.9
Severe	2	5.6
HAV (left foot)		
Mild	7	19.4
Moderate	5	13.9
Severe	0	0
Presented painful points	27	75.0
Forefoot	19	52.8
Rearfoot	13	36.1
Midfoot	3	8.3
Ankle	12	33.3
Beighton Scale (Mean ± SD)	6.4 ± 2.2	
Foot posture index, right foot (Mean ± SD)	5.3 ± 3.0	
Foot posture index (left foot) (Mean ± SD)	5.1 ± 3.2	

HAV: Hallux abducto-valgus.

**Table 4 ijerph-17-01359-t004:** Intervention outcomes.

	Pre			Post			Means Difference	95%CI for Means Difference	*p*	SE
	Mean	SD	Median	Mean	SD	Median				
11-NRS	6.03	2.53	6	4.33	2.78	4.50	1.69	0.77–2.62	0.002 **	0.53
Days with pain	5.78	2.14	7	3.08	2.44	2.50	2.44	1.33–3.64	0.002 **	0.55
FFI-pain	64.68	22.60	29.68	42.36	28.28	45	22.31	14.21–30.42	<0.001 **	0.74
FFI-disability	51.57	27.87	49.44	37.18	28.35	35	14.39	6.58–22.20	0.001 **	0.55
FFI-activity limitation	26.76	27.91	20	12.28	15.53	6	14.48	7.02–21.94	0.001 **	0.55
TOTAL FFI	52.02	22.28	51.16	34.20	24.21	38.90	17.82	11.24–24.40	<0.001 **	0.69
FSS	45.19	12.45	51	40.97	14.47	45.50	4.22	1.19–7.25	0.007 **	0.45
SF-12 physical	26.93	10.02	23.60	26.83	10.59	25.85	0.10	−3.24–3.43	0.649	-
SP-12 mental	33.64	10.39	32.85	38.26	15.81	38.55	-4.62	0.47–−1.84	0.016 *	0.40
MFPDI	22.55	8.33	23	18.53	8.83	20	4.02	1.87–6.20	<0.001	0.62

Mean, standard deviation (SD), median, and inter-quartile ranges (IQR), at the initial measurement and after the 3-month follow-up period, with *p* values (and size effect (SE) only in statistically significant comparisons). 11-NRS: 11-point numeric rating score; FFI: Foot function index; MFPDI: Manchester foot pain and disability index; FSS: Fatigue severity score. * *p* < 0.05; ** *p* < 0.01.

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
