# Peer review of "Custom-Made Foot Orthoses Reduce Pain and Fatigue in Patients with Ehlers-Danlos Syndrome. A Pilot Study"

_ijerph, 2020, doi:10.3390/ijerph17041359_

Round 1

Reviewer 1 Report

This paper is interesting and new way of presenting how to manage EDS problems. So much research is needed to help EDS patients to manage daily life. Congratulations to this paper!

However, some issues are present. This is mostly the presentation of tables.

Why are they so compact? This makes them not easy to read.

Table 1: Percentage and N have moved from their places? Whose familiar history is marked, the participants or their family? What is noxious habits, no figure there? The second part of the table is on next page, makes it difficult to read. 

Table 2: Make it clearer, N and Mean in the same row make it difficult to read. Explain the indexes (Foot posture, Clarke, Arch). 

Table 3: is divided in two pages, difficult to read.

If a picture or scetch of the orthotic could be found in the paperthis would improve the presentation. And it would be more useful to podiatirics that want to help their patients. 

Author Response

Thanks for your commentaries in order to improve the quality of the manuscript. A deep and substantial modification has been carried out according to your suggestions.

In response to: “However, some issues are present. This is mostly the presentation of tables. Why are they so compact? This makes them not easy to read.”: We have modified it.

In response to: Table 1: Percentage and N have moved from their places? Whose familiar history is marked, the participants or their family? What is noxious habits, no figure there? The second part of the table is on next page, makes it difficult to read. We have modified it. The table 1 have been divided into two tables.

In response to: Table 2: Make it clearer, N and Mean in the same row make it difficult to read. Explain the indexes (Foot posture, Clarke, Arch). We have cleared the presentation of table 2.

In response to: Table 3: is divided in two pages, difficult to read. We have corrected it and now this table is only in a page.

In response to: “If a picture or scetch of the orthotic could be found in the pape rthis would improve the presentation. And it would be more useful to podiatirics that want to help their patients.” A picture of the foot orthotics has been included.

Reviewer 2 Report

1) This study highlights an important aspect: finding solutions to improve the living conditions of people with EDS. Quoting the authors: “The EDS is recognised as a severely disabling disease, with chronic pain and fatigue as important 223 factors related to severe deterioration of quality of life. Clinical studies are necessary to improve the general understanding of EDS and help symptoms management and disability prevention”. The work is well structured and the statistical procedures well explained and adequate.

It deserves to be published mainly because it demonstrates and explains a methodology that can be replicated in a clinical situation. The resources applied can be used by clinicians on the area if (as the authors) they customize the insoles. However the insoles customization is well described.

=======

2) FFI was used for “Foot functionality”. FFI is a self-report that proposes to measure the pain and the disability. This Index is used over two decades. FFI, have a wide literature of support and is very convenient to use in situations like the one the authors study had developed (utilization of insoles during 3 months, 7 hours a day). In the present study it´s assumed that the FFI is validated to the Spanish population. Yes/No ? I propose the authors pay attention this point.

=======

3) Table 1. (Participants’ characteristics and comorbidities) and Table 2.( Participants’ foot problems) shows a “normal no homogenous sample” in relation to the health conditions.

Those tables show a “no normal homogenous sample” due the impossibility to find homogeneity in patients sample with EDS. Other issues could be mention. For instance “We did not take into account the potential effect of pain medication”. Please comment this issues in relation with the results.

=======

4) The authors also collected, The Beighton score, a popular screening technique for hypermobility; Manchester scale, that scores how pronounced is the hallux valgus;

Numeric Pain Rating Scale, subjective measure of pain rating.

Quoting: 143 to 145: “The observed improvement in disability from 49.4 points to 35 points measured in FFI, together with the improvement in functional limitation from 20 points to 6 in FFI and from 14 to 11 in MFPDI, shows an improvement in patient mobility after 3 months with the orthosis.”

All of those data are obtained based on scale values. Some scales observed by specialists, others by the sample members themselves. What is the guarantee that these values can be compared with objective data obtained by laboratory means?

Comment: Laboratory results are increasingly common, based on 3D models of movement associated with data on the reactive force of the support or on simple pressure plates.

This point highlights the dichotomy of the clinician's work (necessarily limited to the consultation time) and the data on which this clinician is based to have an intervention based on scientific evidence. These latest data are the responsibility of scientists.

 =======

5) In “Table 2. Participants’ foot problems” the authors separate data from the right foot / left foot. However the final data do not mention this separation explicitly. Please justify.

=======

References sources:

Proposal, look for:

Peterson B, Coda A, Pacey V, Hawke F. Physical and mechanical therapies for lower limb symptoms in children with Hypermobility Spectrum Disorder and Hypermobile Ehlers-Danlos Syndrome: a systematic review. J Foot Ankle Res. 2018 Nov 7;11:59. PMID: 30455744

Following the methodological “school” of Maria Gali (politecnico de Milano) as you mention on

Galli, M.; Cimolin, V.; Vismara, L.; Grugni, G.; Camerota, F.; Celletti, C.; Albertini, G.; Rigoldi, C.;

Capodaglio, P. The effects of muscle hypotonia and weakness on balance: A study on Prader-Willi and Ehlers-Danlos syndrome patients. Res. Dev. Disabil. 2011, 32, 1117–1121

Proposal, look for:

Cimolin V, Galli M, Celletti C, Pau M, Castori M, Morico G, Albertini G, Camerota F. Foot type analysis based on electronic pedobarography data in individuals with joint hypermobility syndrome/Ehlers-Danlos syndrome hypermobility type during upright standing. J Am Podiatr Med Assoc. 2014;104(6):588-93. PMID: 25514270

Author Response

Thanks for your commentaries in order to improve the quality of the manuscript. A deep and substantial modification has been carried out according to your suggestions.

In response to: “FFI was used for “Foot functionality”. FFI is a self-report that proposes to measure the pain and the disability. This Index is used over two decades. FFI, have a wide literature of support and is very convenient to use in situations like the one the authors study had developed (utilization of insoles during 3 months, 7 hours a day). In the present study it´s assumed that the FFI is validated to the Spanish population. Yes/No ? I propose the authors pay attention this point. Yes, the FFI is validated to the Spanish population. We have clarified.

In response to: “Table 1. (Participants’ characteristics and comorbidities) and Table 2.( Participants’ foot problems) shows a “normal no homogenous sample” in relation to the health conditions. Those tables show a “no normal homogenous sample” due the impossibility to find homogeneity in patients sample with EDS. Other issues could be mention. For instance “We did not take into account the potential effect of pain medication”. Please comment this issues in relation with the results. Tables have been modified. Comments regarding the potential effect of pain medication are included in lines 288 to 293.

In response to:

The authors also collected, The Beighton score, a popular screening technique for hypermobility; Manchester scale, that scores how pronounced is the hallux valgus;

Numeric Pain Rating Scale, subjective measure of pain rating.

Quoting: 143 to 145: “The observed improvement in disability from 49.4 points to 35 points measured in FFI, together with the improvement in functional limitation from 20 points to 6 in FFI and from 14 to 11 in MFPDI, shows an improvement in patient mobility after 3 months with the orthosis.”

All of those data are obtained based on scale values. Some scales observed by specialists, others by the sample members themselves. What is the guarantee that these values can be compared with objective data obtained by laboratory means?

Comment: Laboratory results are increasingly common, based on 3D models of movement associated with data on the reactive force of the support or on simple pressure plates.

This point highlights the dichotomy of the clinician's work (necessarily limited to the consultation time) and the data on which this clinician is based to have an intervention based on scientific evidence. These latest data are the responsibility of scientists.

Authors are sorry for not totally understanding this comment. The scales used in this study were administered by a member of the research team talking to the participants, with them at the office (first date), or by phone calls (after follow-up period).

In response to: In “Table 2. Participants’ foot problems” the authors separate data from the right foot / left foot. However the final data do not mention this separation explicitly. Please justify.

There is a justification in Materials and Methods (line 92 and 93).

In response to:

References sources:

Proposal, look for:

Peterson B, Coda A, Pacey V, Hawke F. Physical and mechanical therapies for lower limb symptoms in children with Hypermobility Spectrum Disorder and Hypermobile Ehlers-Danlos Syndrome: a systematic review. J Foot Ankle Res. 2018 Nov 7;11:59. PMID: 30455744

Following the methodological “school” of Maria Gali (politecnico de Milano) as you mention on

Galli, M.; Cimolin, V.; Vismara, L.; Grugni, G.; Camerota, F.; Celletti, C.; Albertini, G.; Rigoldi, C.;

Capodaglio, P. The effects of muscle hypotonia and weakness on balance: A study on Prader-Willi and Ehlers-Danlos syndrome patients. Res. Dev. Disabil. 2011, 32, 1117–1121

Proposal, look for:

Cimolin V, Galli M, Celletti C, Pau M, Castori M, Morico G, Albertini G, Camerota F. Foot type analysis based on electronic pedobarography data in individuals with joint hypermobility syndrome/Ehlers-Danlos syndrome hypermobility type during upright standing. J Am Podiatr Med Assoc. 2014;104(6):588-93. PMID: 25514270

                -The authors thank the reviewer for their suggestions. These two references have been incorporated to the manuscript.

Reviewer 3 Report

Custom-made Foot Orthoses Reduce Pain and Fatigue in Patients with Ehlers-Danlos syndrome

Thank you for the opportunity to undertake a review of this study.  I agree that this is a necessary area for research, with podiatric, in particular the use of foot orthoses in lower-limb management is under-represented in the literature.  However, the study itself is of a weak study design, with multiple measures; there is not insufficient reporting nor control of the intervention or of other management strategies.  I fear that this means findings cannot be suitably considered to be meaningful or trustworthy.

In particular, I think a lot of important exclusion criteria have been missed, for example those already wearing orthoses, previous foot surgery, those who have recently commenced pain medication (or are even on regular pain management medication) etc. 

Obvious other weaknesses alluded to in the study design include a lack of blinding to participants or researchers, no randomisation, nor is there sufficient reporting information on what other management participants are receiving.

we do not know what other management the participants have received in conjunction with the orthoses.

The flow and grammar of the article could be improved for flow and interpretation.  There is also superfluous text which affects readability.  Sentences need to be clear and succinct.

Abstract

The numbers 1), 2) and 3) etc. are unneccessary

Line 16 – ‘The feet may be affected’ can be removed

Line 16 – 17 – I am not sure that the aim is to ‘describe the effects…’, rather to measure change

Line 23 – ‘who’ can be removed

Introduction

More information about EDS could be provided.  Including age, prognosis, morbidity etc.

Line 35 – 39 – The outline of EDS and it’s sub-types is very superficial.  For example, you state that the ‘hypermobile’ type is the most common, but then do not specifically link this to the purpose of your study (i.e. foot pain etc).  This leaves us wondering if this is or is not an issue in these 90% of cases.

Line 36 – ‘was related to various of them’ doesn’t make sense

Line 49 – ‘Treating EDS-related foot pathology…’

Line 44 – 48 – Doesn’t really provide us with a lot of detail to the foot pain or complications which exist in EDS.  This can provide much more detailed information.  Where is the pain, what is it related to, how is gait altered, what are the structural changes etc. etc.

Line 51 – ‘Single case design’ rather than ‘single patients’

Line 53 – 54 – outlining the lack of evidence is repetitive

Materials and Methods

It would be good if you could separate out the tools/outcomes into sub-heading and address these in detail under the headings.  I.e. pain measures, functional measures etc.

Line 58 – I am not familiar with the term ‘non-controlled’ clinical trial.  Does this mean it is merely an observational study.  Particularly since there is also no randomisation, blinding etc. my understanding is everyone is receiving the same treatment. 

Line 60 – 61 – ‘received a favourable evaluation’ is superfluous, if it hadn’t then you wouldn’t have been allowed to undertake the study

Line 74 – 76 – This

Line 76 – Should read ‘independent gait’ rather than ‘autonomous gait’

Line 77 – 78 – ‘years since diagnosis’…

Line 80 – 86 – Insufficient information regarding the orthoses and how they were prescribed is provided.  How they were customised etc. will have a big impact on their function or possible effect.  Was there also a change/evaluation of footwear as the orthoses/footwear interaction is quite important.  Were they advised to wear in slowly?  What was the timeline between the initial appointment and dispense? 

Line 83 – it seems strange with the importance of frequency of use, that there wasn’t a measure of this.  Can you be certain that they were wearing them just by making a phone call?  I am also interested to know, given the regular contact how there were quite so many participants lost to follow-up.  Was this identified earlier during the phone calls.  Also, what if there were adverse events relating to the orthoses?

Line 112 – ‘continuous’ rather than scalar

Results

I would be interested why there were participants lost to follow-up.  It is possible (likely) that these found no benefit to orthoses.  This may have been able to be addressed stastically or through phone call follow-up even?

Tables – could be improved with the use of appropriate headings bolded etc (i.e. Noxious habits/ Beighton Scale etc).  There is also misalignment of columns in table 1 and 2.

What does ‘disability recognized by government’ (table 1) even relate to, and why is it so low?  This hasn’t been addressed in the methods.  Why are the figures for this reversed (i.e. N and % have changed columns)?

All tables need the abbreviations/key provided underneath. 

It would be good to provide tables relevant to the tools used and how they scored for them (and individual elements) as this will also be very important to what you are looking at. 

Line 132 and line 145 – ‘orthoses’ rather than ‘orthosis’

Table 3 lacks a heading

Table 3 – actual P-values should be provided.  Significant values should be bolded and used * < 0.05 ** < 0.001 etc.

Discussion

There has not been any discussion or detail provided around the effect (or possible effect) of orthoses.  For example, in the literature there is plenty of information (even relating to EDS) of proprioceptive and other possible effects of orthoses (such as desensitisation).

Why do you think they have improved things?

There has and needs to be a lot of discussion around the weaknesses of the study… but still many have been missed… including things like, has there been an associated decrease in activity levels (due to the orthoses or even not), has there been a change in footwear to accommodate the orthoses, therefore also having a benefit not specific to the orthosis itself. 

Line 215- 217 – This lacks context and therefore seem superfluous.  What are you trying to say about these studies other than maybe reference other papers?

Author Response

Thanks for your commentaries in order to improve the quality of the manuscript. A deep and substantial modification has been carried out according to your suggestions.

In response to “Thank you for the opportunity to undertake a review of this study. I agree that this is a necessary area for research, with podiatric, in particular the use of foot orthoses in lower-limb management is under-represented in the literature.  However, the study itself is of a weak study design, with multiple measures; there is not insufficient reporting nor control of the intervention or of other management strategies.  I fear that this means findings cannot be suitably considered to be meaningful or trustworthy.”:

The authors agree with the reviewer, as this study would be more rigorous if a control group had been included. Due to the difficulty to recruit individuals with EDS in an adequate number, this pilot study was made without control group. This has been commented in the limitations section.

In response to “In particular, I think a lot of important exclusion criteria have been missed, for example those already wearing orthoses, previous foot surgery, those who have recently commenced pain medication (or are even on regular pain management medication) etc.”:

Obviously these were real exclusion criteria. Actually the participants were asked about them before they were proposed to take part in the study. We want to thank the reviewer for this suggestion. These exclusion criteria that had already been taken into account, have now been included in the text.

In response to “Obvious other weaknesses alluded to in the study design include a lack of blinding to participants or researchers, no randomisation, nor is there sufficient reporting information on what other management participants are receiving. We do not know what other management the participants have received in conjunction with the orthoses.” Certainly there was a lack of blinding to participants or researchers, and no randomization was carried out, because there was no control group. It has been mentioned in the limitations section. In spite of this, the authors think that due to the lack of knowledge regarding the effect of foot orthoses on individuals with EDS, this study could be the initial point for future large controlled clinical trials, as it is the first study that would deal with foot orthosis in EDS.

Reviewer 4 Report

This paper is well written and covers a very interesting pathology with the response to foot orthoses. It may be of value to readers for a visual presentation of the devices used in the study and clarity on if the participants all received the same prescription on the device and if so why? It is common clinical practice in podiatry to customise individual needs for orthoses rather than provide a generic one size fits all.

There are a number of English and format errors that the editorial proof reader may wish to change in line with the journals guide for authors.  

Author Response

Thanks for your commentaries in order to improve the quality of the manuscript. A deep and substantial modification has been carried out according to your suggestions.

In response to: “It may be of value to readers for a visual presentation of the devices used in the study and clarity on if the participants all received the same prescription on the device and if so why? It is common clinical practice in podiatry to customise individual needs for orthoses rather than provide a generic one size fits all.

A picture of the foot orthoses has been included. Although the materials were the same for all participants, weight-bearing phenolic foam molds of the feet were obtained to make the customisation of the foot orthoses, as indicated in the Methods section.

In response to “There are a number of English and format errors that the editorial proof reader may wish to change in line with the journals guide for authors

The flow and grammar of the article could be improved for flow and interpretation.  There is also superfluous text which affects readability.  Sentences need to be clear and succinct.”

A native English spoken professional translator reviewed the language. Please find attached the certificate he delivered. Should another revision be necessary, authors will send the manuscript for new edition.

In response to: “Abstract The numbers 1), 2) and 3) etc. are unnecessary”

The numbers have been removed.

In response to: “Line 16 – ‘The feet may be affected’ can be removed”

It has been removed.

In response to: “Line 16 – 17 – I am not sure that the aim is to ‘describe the effects…’, rather to measure change”

It has been changed.

In response to: “Line 23 – ‘who’ can be removed”

It has been removed. In response to:

“Introduction

More information about EDS could be provided.  Including age, prognosis, morbidity etc.

Line 35 – 39 – The outline of EDS and it’s sub-types is very superficial.  For example, you state that the ‘hypermobile’ type is the most common, but then do not specifically link this to the purpose of your study (i.e. foot pain etc).  This leaves us wondering if this is or is not an issue in these 90% of cases.”

The authors have not considered necessary to go deeper into more information about the EDS to respect an adequate number of words in the manuscript. As this information is not original to this research, the authors have intended to provide more text related to the original findings of their study. However, if it is considered necessary to extend the introduction, the authors will be happy to do so.

In response to: “Line 36 – ‘was related to various of them’ doesn’t make sense”

It has been change by ‘was present in several of them’.

In response to: “Line 44 – 48 – Doesn’t really provide us with a lot of detail to the foot pain or complications which exist in EDS. This can provide much more detailed information.  Where is the pain, what is it related to, how is gait altered, what are the structural changes etc. etc.”

Some more information has been added in these lines.

In response to: “Line 51 – ‘Single case design’ rather than ‘single patients’

It has been changed.

In response to: “Line 53 – 54 – outlining the lack of evidence is repetitive”

The sentence has been modified.

In response to: “Materials and Methods. It would be good if you could separate out the tools/outcomes into sub-heading and address these in detail under the headings. e. pain measures, functional measures etc.

Sub-headings have been added.

In response to: “Line 58 – I am not familiar with the term ‘non-controlled’ clinical trial. Does this mean it is merely an observational study.  Particularly since there is also no randomisation, blinding etc. my understanding is everyone is receiving the same treatment.” The type of study has been changed.

In response to: “Line 60 – 61 – ‘received a favourable evaluation’ is superfluous, if it hadn’t then you wouldn’t have been allowed to undertake the study.”

It has been deleted.

In response to: “Line 76 – Should read ‘independent gait’ rather than ‘autonomous gait’”

It has been modified.

In response to: “Line 77 – 78 – ‘years since diagnosis’…”

It has been changed.

In response to: “Line 80 – 86 – Insufficient information regarding the orthoses and how they were prescribed is provided. How they were customised etc. will have a big impact on their function or possible effect.  Was there also a change/evaluation of footwear as the orthoses/footwear interaction is quite important.  Were they advised to wear in slowly?  What was the timeline between the initial appointment and dispense?”

Some more information has been added regarding the instructions given to the participants.

In response to: “Line 83 – it seems strange with the importance of frequency of use, that there wasn’t a measure of this. Can you be certain that they were wearing them just by making a phone call?  I am also interested to know, given the regular contact how there were quite so many participants lost to follow-up.  Was this identified earlier during the phone calls.  Also, what if there were adverse events relating to the orthoses?” Phone calls were made once a month during the follow up period to ensure that people were following the instruction given at the beginning of the study, to ask whether they needed some additional adjustment of the foot orthoses, and to record how many hours per week they were wearing the orthoses. As the researchers verified that the instructions were being followed (at least that was what participants sustained), these data have not been included in the statistical analysis. People who said that he/she did not followed the initial indications were excluded.

In response to: “Line 112 – ‘continuous’ rather than scalar”

It has been modified.

In response to: “Results. I would be interested why there were participants lost to follow-up. It is possible (likely) that these found no benefit to orthoses.  This may have been able to be addressed stastically or through phone call follow-up even?”

It has been modified (line 152-54).

In response to: “Tables – could be improved with the use of appropriate headings bolded etc (i.e. Noxious habits/ Beighton Scale etc). There is also misalignment of columns in table 1 and 2.”

The tables have been modified.

In response to: “What does ‘disability recognized by government’ (table 1) even relate to, and why is it so low? This hasn’t been addressed in the methods.  Why are the figures for this reversed (i.e. N and % have changed columns)?”

Table where this appears has been modified for a better understanding.

In response to: “All tables need the abbreviations/key provided underneath”

It has been provided.

In response to: “It would be good to provide tables relevant to the tools used and how they scored for them (and individual elements) as this will also be very important to what you are looking at.”

Thanks for the suggestion. With the modifications made, the manuscript has 4 tables and 1 photo. We think this is an adequate number of tables/figures. In addition, table 4 provides information obtained by means of the assessment tools used.

In response to: “Line 132 and line 145 – ‘orthoses’ rather than ‘orthosis’”

It has been modified.

In response to: “Table 3 lacks a heading”

A heading has been added.

In response to: “Table 3 – actual P-values should be provided. Significant values should be bolded and used * < 0.05 ** < 0.001 etc.

Significant values have been bolded and *<0.05 and **<0.01 has been used.

In response to: “Discussion. There has not been any discussion or detail provided around the effect (or possible effect) of orthoses. For example, in the literature there is plenty of information (even relating to EDS) of proprioceptive and other possible effects of orthoses (such as desensitisation). Why do you think they have improved things?

It have been modified (line 220, 221, 227 and 229).

In response to: There has and needs to be a lot of discussion around the weaknesses of the study… but still many have been missed… including things like, has there been an associated decrease in activity levels (due to the orthoses or even not), has there been a change in footwear to accommodate the orthoses, therefore also having a benefit not specific to the orthosis itself.

Main limitations have been discussed in lines 277 to 293. Some modifications made as, for example, more clarification of the exclusion criteria or instruction given to the participants regarding the foot wear, could have clarify that the results must be interpreted with caution.

In response to: Line 215- 217 – This lacks context and therefore seem superfluous. What are you trying to say about these studies other than maybe reference other papers?

We do not completely understand this suggestion, as lines 215 to 217 do not contain any references to other studies. Should some additional modification in this regard is necessary, please indicate and we would be delighted to carry it out.